# Chiral adiabatic transmission protected by Fermi surface topology

Isidora Araya Day[1, 2*], Kostas Vilkelis[1,2], Antonio L. R. Manesco[2],
Ahmet Mert Bozkurt[1, 2†], Valla Fatemi[3] and Anton R. Akhmerov[2‡]

**1** QuTech, Delft University of Technology, Delft 2600 GA, The Netherlands
**2** Kavli Institute of Nanoscience, Delft University of Technology,
P.O. Box 4056, 2600 GA Delft, The Netherlands
**3** School of Applied and Engineering Physics,
Cornell University, Ithaca, NY 14853 USA

⋆ iarayaday@gmail.com , † a.mertbozkurt@gmail.com , ‡ cat@antonakhmerov.org

## Abstract

We demonstrate that Andreev modes that propagate along a transparent Josephson junction have a perfect transmission at the point where three junctions meet. The chirality and the number of quantized transmission channels is determined by the topology of the Fermi surface and the vorticity of the superconducting phase differences at the trijunction. We explain this chiral adiabatic transmission (CAT) as a consequence of the adiabatic evolution of the scattering modes both in momentum and real space. The dispersion relation of the junction then separates the scattering trajectories by introducing inaccesible regions of phase space. We expect that CAT is observable in nonlocal conductance and thermal transport measurements. Furthermore, because it does not rely on particle-hole symmetry, CAT is also possible to observe directly in metamaterials.

---

Unlike particles that follow deterministic trajectories, waves, both quantum and classical, may split and follow multiple paths. Under special conditions, however, waves follow a deterministic path, transmitting perfectly from source to target. The simplest mechanism that protects such transmission is the adiabaticity of the potential landscape: if the potential changes slowly enough, the wave functions adjust to the local changes of the potential without splitting into partial waves. In a quantum point contact [1], for example, the adiabaticity of the constriction ensures that an integer number of modes pass through, while the rest of the modes reflect. Another mechanism that protects quantized transmission is the topology of a gapped Hamiltonian, which prohibits scattering between channels due to a combination of their symmetry structure and spatial separation. For example, the chiral edge transport of a quantum Hall insulator [2] is protected because the channels propagating in opposite directions occupy different edges of the sample, and are separated by a gapped bulk.

Topological protection, however, extends beyond the bulk properties of an insulator. Specifically, the number of electron- and hole-like Fermi surfaces give rise to the quantized transmission of Andreev modes propagating in a superconductor – normal metal – superconductor (SNS) junction  at a $\pi$ phase difference [3]. While these modes are dispersionless

within the Andreev approximation (the linearization of the Hamiltonian at the Fermi level) they acquire charge and velocity due to the nonlinearity of the normal dispersion. At positive voltage bias, the nonlocal conductance measures the number of electron-like *critical points*: Fermi surface points where the velocity is parallel to the interface between the superconductors. Likewise, negative bias conductance counts the number of hole-like critical points. The difference between electron and hole-like critical points is the Euler characteristic of the Fermi surface—a topological invariant [4].

To highlight our main result, we refer the reader to Fig. 1: a multiterminal short SNS junction has quantized chiral transmission of Andreev modes (see App. A and Ref. [5] for the details of numerical simulations). In this device, pairs of superconductors form waveguides for Andreev modes, which occupy an energy range below the superconducting gap and above a minimal energy determined by the phase difference across the junction. At the point where multiple junctions meet, the Andreev modes from different waveguides perfectly transmit clockwise or counterclockwise, depending on the winding of the superconducting phases.

While protected chiral transport also exists in quantum Hall systems, the modes in the SNS junction occupy the same spatial region, and therefore the mechanism is distinct. Furthermore, the different scattering trajectories are not distinguished by any quantum number, which excludes symmetry-based protection. In the following, we examine this phenomenon and explain how chiral transmission emerges from the dispersion of the Andreev states beyond the Andreev approximation and the topology of the Fermi surface. Because the scattering is protected by the adiabaticity of the wave function evolution, we name this phenomenon *chiral adiabatic transmission* (CAT).

To understand the origin of CAT, we consider a Josephson junction: a normal metal between two superconductors, as shown in Fig. 2(a). We generalize the result of Ref. [3] to an arbitrary phase difference between the superconductors, as necessary to analyze a trijunction. To identify the role of corrections to the Andreev approximation, we consider a parabolic dispersion in the direction $y$, perpendicular to the interface between the superconductors.

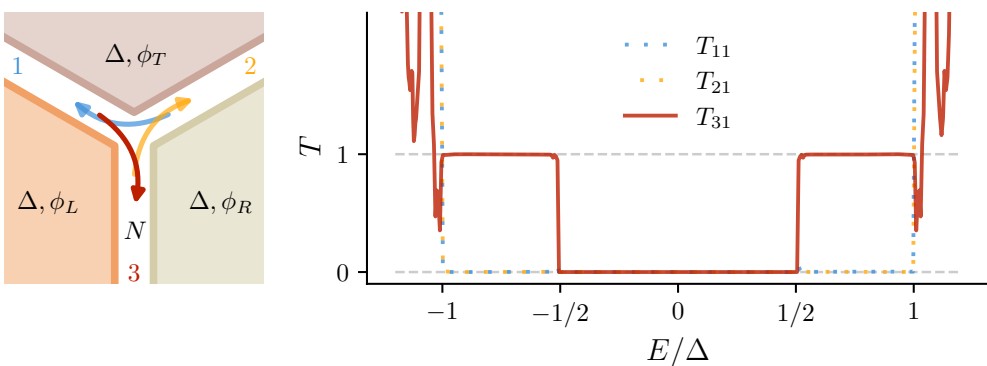

Figure 1: A three-terminal Josephson junction has quantized chiral transmission of Andreev modes. Left: Three superconductors with an infinitesimally narrow ballistic metal in between ($N$). The superconductors have the same normal Hamiltonian with chemical potential $\mu$. The gap $\Delta$ is constant, but the superconducting phases $\phi_T$, $\phi_L$, and $\phi_R$ differ, such that all the phase differences are $2\pi/3$. Andreev modes propagate along the junctions as shown by the arrows. Right: Transmission of Andreev modes from lead 1 into itself ($T_{11}$), lead 2 ($T_{21}$), and lead 3 ($T_{31}$), for a Y-shaped three-fold symmetric junction. Transmissions from leads 2 and 3 are not shown, but are likewise quantized and chiral. Above the superconducting gap the scattering modes are not confined to the junctions and all transmission become enabled.

This is a good approximation close to a critical point and still gives a qualitatively valid description of the dispersion away from the critical point. We consider two s-wave superconductors with a gap $\Delta$ and a phase difference $\delta\phi$, with an infinitesimally narrow metal between them. At a fixed momentum $k_x$ parallel to the junction, the Bogoliubov-de Gennes Hamiltonian reads:

$$H(y) = [-a\partial_y^2 - E_x]\tau_z + \Delta\cos(\delta\phi/2)\tau_x + \text{sign}(y)\Delta\sin(\delta\phi/2)\tau_y, \tag{1}$$

where $\tau_{x,y,z}$ are Pauli matrices acting on the particle-hole degree of freedom, $2a$ is the inverse effective mass, and $E_x$ is the position of the band bottom. Considering the dispersion near a critical point with $k_x = k_c + \delta k_x$ gives $E_x = -v_x\delta k_x$, and reproduces the Hamiltonian of Ref. [3] when $\delta\phi = \pi$. For a parabolic band, on the other hand, $E_x = \mu - ak_x^2$, with $\mu$ the chemical potential.

The Andreev approximation follows from linearizing the dispersion around the Fermi momentum $k_{y0} = \pm\sqrt{E_x/|a|}$ in the $y$-direction and using the Ansatz $|\Psi(y)\rangle = \exp(ik_{y0}y)|\psi\rangle$, where $|\psi\rangle$ is a two-component spinor that only changes slowly with $y$. This approximation is valid when $\Delta \ll E_x$. Applying Eq. (1) to $|\Psi(y)\rangle$ and neglecting $\partial_y^2|\psi\rangle$, we find that $|\psi\rangle$ is an eigenstate of the linearized Hamiltonian

$$H_\pm^{(0)}(y) = \mp 2iak_{y0}\partial_y\tau_z + \Delta\cos(\delta\phi/2)\tau_x + \text{sign}(y)\Delta\sin(\delta\phi/2)\tau_y. \tag{2}$$

This Hamiltonian has one bound state for each sign of $\pm k_{y0}$, which we use to construct the approximate eigenstates of $H(y)$:

$$|\Psi_\pm^{(0)}(y)\rangle = \sqrt{\frac{\Delta\sin|\delta\phi/2|}{v_y}}\begin{pmatrix}\pm 1\\1\end{pmatrix}\exp\left[\pm ik_{y0}y - \frac{\Delta\sin|\delta\phi/2|}{v_y}|y|\right], \tag{3}$$

where $v_y = 2ak_{y0}$. The corresponding eigenvalues $E_\pm^{(0)} = \pm\Delta\cos(\delta\phi/2)$ are the result of the Andreev approximation. To go beyond the linear approximation, we project the full Hamiltonian $H(y)$ onto the basis states $|\Psi_\pm^{(0)}(y)\rangle$, keep only terms up to $\mathcal{O}(\Delta^2)$, and obtain the effective Hamiltonian

$$H_\pm = \Delta\begin{pmatrix}\cos(\delta\phi/2) & \frac{\Delta}{2E_x}\sin^2(\delta\phi/2)\\\frac{\Delta}{2E_x}\sin^2(\delta\phi/2) & -\cos(\delta\phi/2)\end{pmatrix}. \tag{4}$$

This yields the dispersion of the Andreev modes

$$E_\pm = \pm\Delta\sqrt{(\Delta/2E_x)^2\sin^4(\delta\phi/2) + \cos^2(\delta\phi/2)}, \tag{5}$$

and the corresponding eigenstates

$$N_\pm|\Psi_\pm(y)\rangle = \left(\cos(\delta\phi/2) \pm \frac{E_\pm}{\Delta}\right)|\Psi_+^{(0)}(y)\rangle + \frac{\Delta}{2E_x}\sin^2(\delta\phi/2)|\Psi_-^{(0)}(y)\rangle, \tag{6}$$

where $N_\pm$ is a normalization factor. The relative weights of the momenta $\pm k_{y0}$ contributed by $|\Psi_\pm^{(0)}(y)\rangle$ depend on $\delta\phi$ and $E_x/\Delta$, and are only equal at $\delta\phi = \pi$. Away from $\delta\phi = \pi$, the Andreev modes are asymmetric superpositions of the states at $\pm k_{y0}$, and the average momentum of the Andreev modes is misaligned with the junction. Figure 2(b) shows the orientation $\theta = \arctan(k_x/\langle k_y\rangle)$ of the average Andreev mode momentum as a function of $k_x$. At the lowest available energy, the momentum of the Andreev modes is perpendicular to the interface ($\theta \in \{0,\pi\}$). Modes with $k_x$ close to the critical value have $\theta \approx \pm\pi/2$, and additionally their energy increases and eventually exceeds the superconducting gap. There are no modes with momentum parallel to the junction.

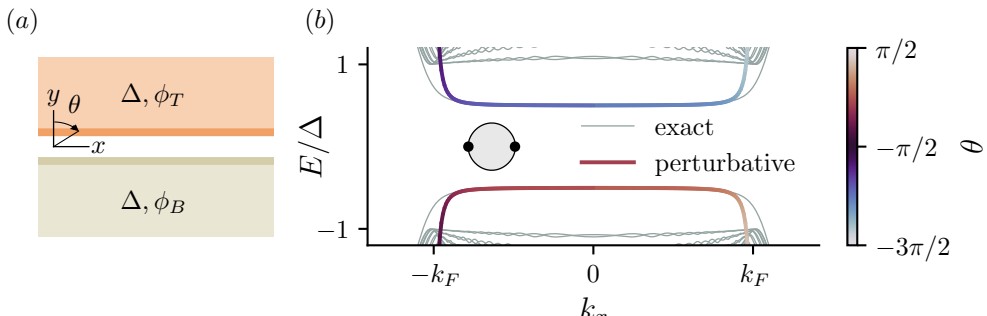

Figure 2: The quadratic dispersion of the Andreev modes hybridize states at opposite momenta, enabling the propagation of the Andreev modes along the junction. (a) SNS short junction with arbitrary $\phi_L - \phi_R$ phase difference. (b) Spectrum of the junction in (a) for a normal dispersion with circular electron Fermi surface, and $\phi_L - \phi_R = 2\pi/3$ computed numerically (gray lines). The inset shows the Fermi surface with the critical points (black dots) of the Fermi surface have $k_x = \pm k_F$, the Fermi wave vector. The perturbative dispersion of the Andreev modes (5) is colored according to the angle of the momentum expectation value $\theta = \arctan\left(k_x/\langle k_y\rangle\right)$.

In the arms of the trijunction the average momentum of the Andreev modes is slightly misaligned with the junction's direction. Near the intersection, the momentum of the scattering states changes continuously because the superconducting pairing—the only position-dependent Hamiltonian term—is small. While some scattering processes at the trijunction may occur without aligning the momentum of the Andreev modes with the arms of the trijunction (see Fig. 3(left)), others necessarily require the momentum to be aligned with the arms at some point during the propagation (see Fig. 3(right)). We therefore observe that the dispersion of the arms of the junction forms prohibited regions in the phase space of the scattering modes. To confirm the phase space separation of different scattering trajectories, we perform the Wigner-Weyl transform of the scattering wave functions in the trijunction:

$$\Phi(\mathbf{r}, \mathbf{k}) = \int d\mathbf{r}'\, e^{-i\mathbf{k}\cdot\mathbf{r}'/\hbar} \psi^*\left(\mathbf{r} + \frac{\mathbf{r}'}{2}\right)\psi\left(\mathbf{r} - \frac{\mathbf{r}'}{2}\right), \tag{7}$$

This transform gives the Wigner quasiprobability distribution $\Phi(\mathbf{r}, \mathbf{k})$ of the wave function $\psi$ in phase space. We use that the Wigner distribution is peaked near the Fermi surface and further simplify it by only considering $\mathbf{k} = (k_F(E)\cos\theta, k_F(E)\sin\theta)$, with $k_F(E)$ the Fermi momentum at the energy of interest. We show in Fig. 4(a) the resulting Wigner distributions $\Phi_i(x, y, \theta)$ of the scattering wave functions $\psi_i$ injected from lead $i$ in the Y-shaped junction shown in Fig. 1, and confirm that $\Phi_i$ indeed do not overlap. Because the asymptotic values of $\theta$ depend on the orientation of the trijunction arms, we also expect that the semiclassical trajectories overlap if we decrease the relative angle between any two arms. We confirm this by computing the Wigner distribution of scattering modes in an arrow-shaped trijunction shown in Fig. 4(b), where the angle between two pairs of arms is acute. In this geometry the separation of the scattering wave functions in phase space is lost, and therefore the transmission of Andreev quasiparticles is no longer quantized, as shown in Fig. 4(c). The separation of modes in momentum space is reminiscent of the mechanism protecting quasi-Majorana modes: approximate zero modes appearing in topologically trivial superconductors with broken time-reversal symmetry in presence of smooth confining potentials [6–8].

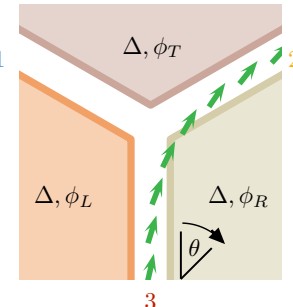 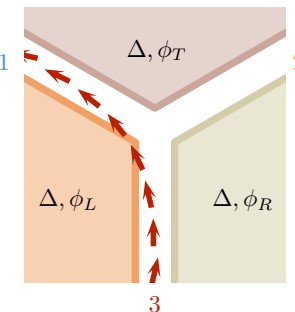

Figure 3: Allowed (left) and prohibited (right) scattering processes at the trijunction. The energy of a mode in each of the three metallic arms of the trijunction is as a function of its orientation $\theta$ and the phase difference $\delta\phi$. The orientation of the arrows depict the momentum of the modes at each position and at an energy that lies within the superconducting gap.

Our arguments rely exclusively on the phase space separation of the scattering wave functions and the adiabatic evolution of the momentum. Therefore, it is natural to expect that CAT does not depend on the details of the junction, the shape of the Fermi surface, or even the presence of particle-hole symmetry. To confirm this assumption, we simulate a trijunction with the following modifications:

- The phase differences across the three junctions are unequal.

- The junction has unequal angles between its arms.

- The Fermi surface is anisotropic.

- Particle-hole symmetry is artificially broken.

The resulting transmissions are shown in Fig. 5 (see Appendix A for the details of the numerical simulations). Despite the modifications, the only qualitative difference from the symmetric trijunction is that the different channels are open at different energy ranges due to the different phase differences. At the energy where modes only exist in one arm, the only allowed process is a reflection of the Andreev modes. At the energy when a conduction channel opens in another arm, the Andreev modes perfectly transmit between the two arms. Finally, only when the three arms have open channels, chiral and quantized transmission is possible and realized.

To prove that CAT is protected by the topology of the Fermi surface, rather than the number of open channels, we consider a model with a next-nearest neighbor hopping in the $y$-direction, such that it has a peanut-shaped Fermi surface. The resulting transport simulations are shown in Fig. 6. The additional critical points of the Fermi surface that appear in two out of three arms of the trijunction create extra particle-like and hole-like channels [3]. At the trijunction the additional channels couple in a way that is sensitive to the junction shape and may either reflect or partially transmit. Despite that, examining the individual transmission eigenvalues—eigenvalues of $t_{ji}t_{ji}^{\dagger}$ with $t_{ji}$ the transmission matrix from lead $i$ to $j$—reveals that one of the eigenvalues stays quantized and chiral. This once again follows from the phase space separation of the scattering modes: at least one of the chiral scattering channels in each arm is separated from other all other modes.

So far we focused on the transmission of Andreev modes, without considering the electrical conductance. Our work differs from Ref. [3] in that we consider finite phase differences, and therefore do not rely on time-reversal symmetry. On the other hand, the electrical conductance in Ref. [3] is quantized because it is impossible to couple opposite sides of the Fermi surface

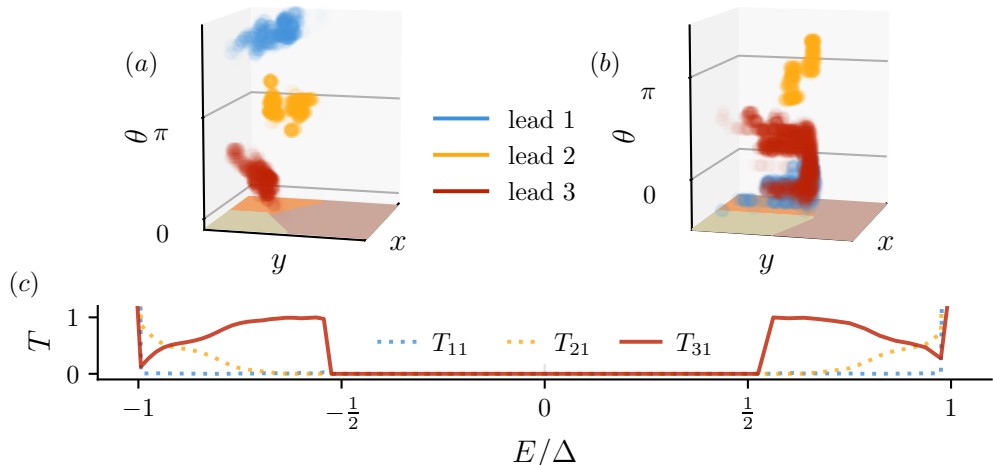

Figure 4: Wigner distribution of the scattering wave functions in two trijunctions with all phase differences equal to $2\pi/3$. For clarity, we only show points where the probability density is above 0.2 times the maximum value. The colors label the leads from which the scattering wave functions are injected. In a Y-shaped junction the scattering wave functions are separated in phase space (a), and their transmission is protected (shown in Fig. 1). In contrast, in a junction with acute angles between the arms, the Wigner distributions of the scattering wave functions overlap (b). The overlap in the phase space destroys the quantization of the transmissions $T_{i1}$ from lead 1 (c). Transmissions from leads 2 and 3 are not shown.

in the absence of large momentum scattering. To confirm the robustness of the electrical conductance quantization at arbitrary phase differences, we simulate the interface between a Josephson junction and a normal lead, and compute the transmissions from the Andreev mode to the electron modes $T_{ea}$ and hole modes $T_{ha}$, with the result being shown in Fig. 7(a). The electrical conductance in a symmetric NSN geometry equals to $(e^2/h)T_{ea}(T_{ea} - T_{ha})$. We observe that similarly to Ref. [3], the Andreev mode perfectly couples to electron modes at positive bias voltage, and hole modes at negative bias voltage, despite breaking time-reversal symmetry. This generalization to arbitrary phase differences can be understood by thinking of the interface as an SNS junction whose metallic region increases in width until it becomes a normal lead, so that each Andreev mode adiabatically evolves into an electron or hole mode. The perfect injection of electrons into Andreev modes followed by chiral adiabatic transmission and perfect emission of Andreev modes into electrons, results in a quantized nonlocal electrical conductance. This enables a purely electric measurement of chiral adiabatic transmission.

The coupling between the two approximate eigenstates $|\Psi_\pm^{(0)}\rangle$ is $\propto \Delta^2/E_x$, and it is similar to the energy $\Delta^2/\mu$ of a Caroli-de Gennes-Matricon (CdGM) bound state in a superconducting vortex [9], where $\mu$ is the distance between the band bottom and the Fermi level. This similarity is not accidental: like the Andreev modes, the momentum distribution of the CdGM states is confined to a cross-section of the Fermi surface and their wave functions possess a similar electron-hole asymmetry. In Fig. 7(b), we show that transmission from a CdGM mode to electrons and holes has the same quantized conductance as that of the Andreev modes. We compute the transmissions between two semi-infinite leads: one a superconductor with a vortex and the other a normal metal. Differently from Andreev modes, however, higher energy CdGM modes contribute the same conductance as the lowest energy mode, so that the total number of quantized conductance channels in a vortex is proportional to $\mu/\Delta$. The

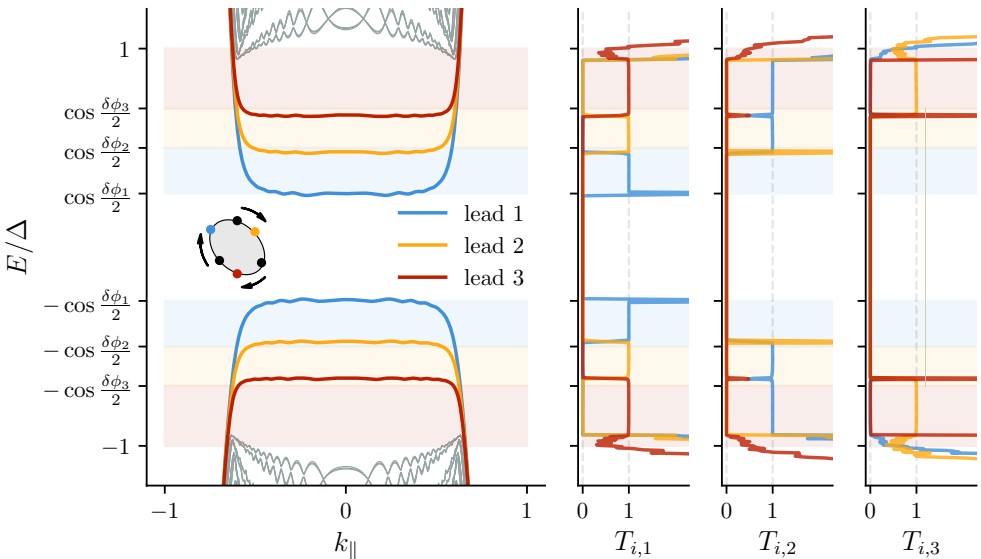

Figure 5: Dispersion and quantized chiral transmission of Andreev modes in an asymmetric trijunction with unequal phase differences $\delta\phi_1, \delta\phi_2, \delta\phi_3$. Left: Dispersion of each lead along the junction's direction. The dispersion of modes confined in the junction is colored according to the lead, and the bulk states' dispersion is shown in grey. The leads have the same normal Hamiltonian with an anisotropic Fermi surface (inset), but the critical points are at different momenta, schematically shown with the dots. Right: Transmission of Andreev modes from leads 1, 2, and 3, respectively, into the other leads or themselves. The colors label the outgoing leads.

conductance quantization of CdGM modes, together with the unexplained quantization of the rectified conductance in a superconducting quantum point contact [4] hints at a more universal description of the underlying protection.

An experimental observation of CAT requires a ballistic Josephson junction. While we considered a position-independent normal Hamiltonian, we expect that a sufficiently smooth potential landscape will not affect the transmission. Thus, candidate platforms must have high mobility and smooth normal-superconductor interfaces, potentially realizable in several platforms. Devices with these properties have been fabricated using two-dimensional electron gases [10–12] and stackings of graphene with superconducting transition metal dicalchogenides [13–15]. Alternatively, twisted bilayer graphene and Bernal bilayer graphene offer gate-defined Josephson junctions with intrinsic superconductivity tunable by electric fields [16–19]. The ability to measure nonlocal electrical conductance while the superconductors are grounded poses an additional challenge to observe the imprint of the Fermi surface topology on the Andreev transport. To suppress the contribution of the supercurrent to nonlocal conductance, many experiments operate in the tunneling regime [10, 20–22], which breaks the quantization of nonlocal conductance. On the other hand, because CAT produces asymmetry of transmission rather than only quantization, it becomes easier to observe. For instance, in addition to purely 2D systems, we expect chiral transport to manifest in high quality films of crystalline superconductors such as aluminum, where the Josephson junctions are formed by narrowing the film thickness. Finally, the chiral nature of the transport makes it observable in thermal transport measurements, which are less sensitive to supercurrent.

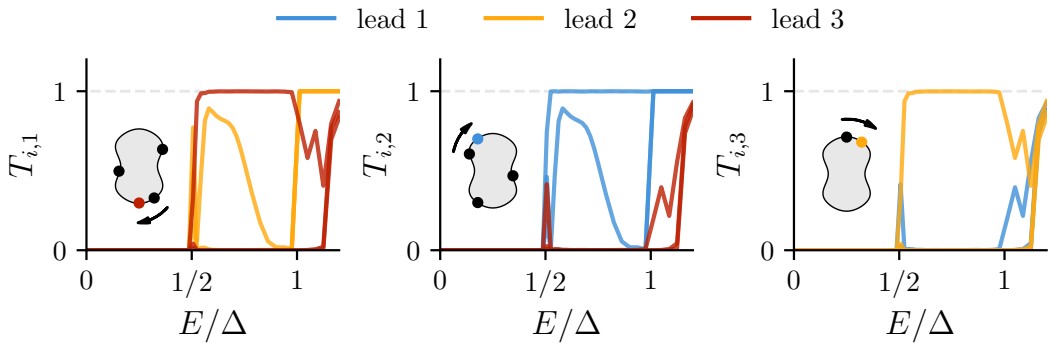

Figure 6: CAT in a trijunction with a peanut-shaped Fermi surface, with Euler characteristic $\chi_F = 1$ equal to that of a circular Fermi surface. The panels show the transmission eigenvalues of Andreev modes from lead 1, 2, and 3, respectively, into the other leads, while reflections are not shown. Only one of the eigenvalues per panel is quantized and chiral, process shown by the arrows from critical points in the incoming lead (black dots) to the critical points in the outgoing lead (colored dots). Unprotected transmissions are those not quantized for $\Delta/2 < E < \Delta$.

Metamaterials offer another platform to observe CAT. Because introducing phase differences requires breaking time-reversal symmetry, single valley transport in photonic or acoustic honeycomb crystals [23–26] is a promising starting point. In such a system coupling an electron-like and a hole-like band that coexist in a single valley mimics the effect of the superconducting pairing. A displacement of the valleys in momentum space then shifts the relative phase difference, implementing an analog of the superconducting phase difference. In addition to microscopic control over the effective Hamiltonian, metamaterials naturally allow local high resolution probes and therefore make the chiral nature of the scattering modes directly observable.

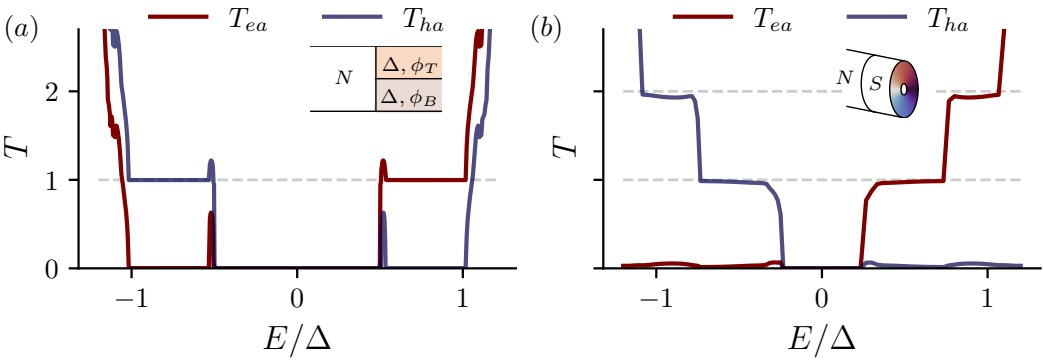

Figure 7: Quantization of coupling between Andreev modes and electrons and holes at a normal metal – Josephson junction interface. (a) Transmission of Andreev modes into electron ($T_{ea}$) and hole ($T_{ha}$) modes from an infinitesimally narrow Josephson junction lead into a metallic one. (b) Transmission of Andreev modes into electron ($T_{ea}$) and hole ($T_{ha}$) modes from a superconducting vortex with a finite metallic core into a metallic lead.

In summary, we have analyzed the transport of Andreev modes in a three terminal Josephson junction. We demonstrated that the Fermi surface topology and adiabaticity enable quantized chiral transmission by separating different channels in momentum space. The chiral nature of the transport makes it observable in thermal, rather than only electrical, transport measurements. Furthermore, because the transmission only relies on the adiabaticity, rather than particle-hole symmetry, this phenomenon is also observable in metamaterials. That the same phenomenology applies to superconducting vortices suggests a more general underlying description, which we leave for future work.

## Acknowledgments

We are grateful to T. Vakhtel for insightful discussions. We thank A. Bordin and A. Young for useful discussions regarding the experimental implementation of the trijunction.

**Data availability**    The code used to produce the reported results and the generated data are available on Zenodo [5].

**Author contributions**    I. A. D., K. V., A. M., and A. A. performed the numerical simulations. I. A. D., K. V., and A. A. prepared the figures. I. A. D., K. V., A. M., and A. A. wrote the manuscript with input from M. B. and V. F. All authors analyzed the results and participated in defining the project scope. A. A. oversaw the project.

**Funding information**    This work was supported by the Netherlands Organization for Scientific Research (NWO/OCW) as part of the Frontiers of Nanoscience program, an NWO VIDI grant 016.Vidi.189.180, and OCENW.GROOT.2019.004. We also acknowledge funding from the European Research Council (ERC) under the European Union's Horizon 2020 research and innovation program grant agreement №828948 (AndQC).

## A   Numerical simulations

The content in the figures of this paper was computed simulating the following tight-binding Hamiltonian using Kwant [27]:

$$H = H_{\text{superconductor}} + H_{\text{normal}}, \tag{A.1}$$

$$H_{\text{superconductor}} = \sum_{\boldsymbol{n}} \left( \Delta e^{i\phi_n} c_{\boldsymbol{n},\uparrow}^{\dagger} c_{\boldsymbol{n},\downarrow}^{\dagger} + \Delta e^{-i\phi_n} c_{\boldsymbol{n},\downarrow} c_{\boldsymbol{n},\uparrow} \right), \tag{A.2}$$

$$H_{\text{normal}} = \sum_{\sigma=\uparrow,\downarrow} \sum_{\boldsymbol{n}} \left[ \left( 2t_x + 2t_y - \mu \right) c_{\boldsymbol{n},\sigma}^{\dagger} c_{\boldsymbol{n},\sigma} - \left( t_x c_{\boldsymbol{n}+\boldsymbol{e}_x,\sigma}^{\dagger} c_{\boldsymbol{n},\sigma} + t_y c_{\boldsymbol{n}+\boldsymbol{e}_y,\sigma}^{\dagger} c_{\boldsymbol{n},\sigma} + \text{h.c.} \right) \right], \tag{A.3}$$

where $c_{\boldsymbol{n},\sigma}^{\dagger}$ ($c_{\boldsymbol{n},\sigma}$) creates (annihilates) an electron with spin $\sigma$ at site $\boldsymbol{n} = (n_x, n_y)$ on a square lattice. The superconducting phase $\phi_{\boldsymbol{n}}$ is site-dependent, while the superconducting gap $\Delta$, chemical potential $\mu \in \mathbb{R}$, and the hopping amplitudes $t_x, t_y$ are uniform.

To compute the transmission in the three-fold symmetric trijunction of Fig. 1 we use a square lattice of size $L = 100$, with parameters $\mu = 0.5$, $\Delta = 0.1$, $t_x = t_y = 1$, and phases $\phi_L = 2\pi/3$, $\phi_R = -2\pi/3$, and $\phi_T = 0$ for the left, right, and top regions respectively. The system for the trijunction is shown in Fig. 8(a), where $\beta = 2\pi/3$ for the three-fold symmetric system. We use the same parameters to compute the band structure in Fig. 2 and the Wigner

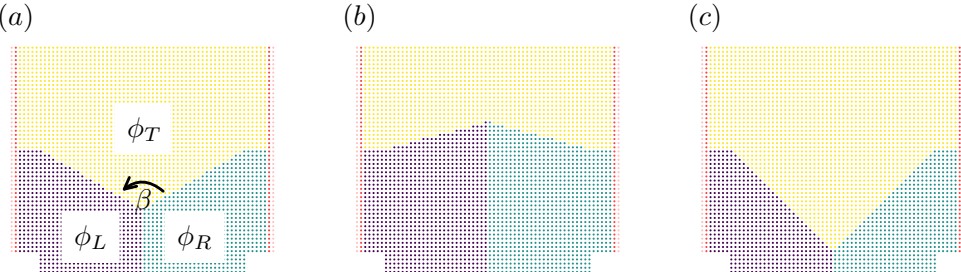

Figure 8: The trijunction geometry. The system is composed of three superconducting regions (left, right, and top) of superconducting phase $\phi_L$, $\phi_R$, and $\phi_T$, respectively. The metallic region between them is not present, because it is infinitesimally thin. The left and right leads (red) are periodic in the x-direction, and the bottom lead (red) is periodic in the y-direction. The angle $\beta$ between the left and right arms of the junction is $2\pi/3$ in (a), $7\pi/6$ in (b), and $\pi/2$ in (c).

distribution in Fig. 4(a). To compute the Wigner distribution in Fig. 4(b) and the wavefunctions' transmissions in Fig. 4(c) we use the same parameters, but we change decrease the angle between the arms such that $\beta = 7\pi/6$, as shown in Fig. 8(b). The code used to compute the transmission is available at [5], together with the code used to generate all the other figures in this paper.

In Fig. 5 we make Fermi surface anisotropic by adding the diagonal hoppings to the Hamiltonian:

$$H_{\text{anisotropy}} = \sum_{\sigma=\uparrow,\downarrow} \sum_{n} \left[ 2t_{xy}c^\dagger_{n,\sigma}c_{n,\sigma} - \left( t_{xy}c^\dagger_{n+e_x+e_y,\sigma}c_{n,\sigma} + \text{h.c.} \right) \right]. \tag{A.4}$$

To compute the spectrum and transmissions in Fig. 5, we use a square lattice of size $L = 100$, with parameters $\mu = 0.5$, $\Delta = 0.1$, $t_x = t_y = 1$, and phases $\phi_L = 2\pi/3 + 1/2$, $\phi_R = -2\pi/3$, and $\phi_T = 0$ for the left, right, and top regions respectively. Moreover, we double the electron block of the Hamiltonian $H_{ee} \to 2H_{ee}$ to artificially break particle-hole symmetry, and we change the angle between the left and right arms of the junction to $\beta = \pi/2$, as shown in Fig. 8(c).

In Fig. 6 we introduce the peanut-shaped Fermi surface by adding the next-nearest neighbor hoppings to the Hamiltonian:

$$H_{\text{peanut}} = -\sum_{\sigma=\uparrow,\downarrow} \sum_{n} \left[ t_{yy}c^\dagger_{n+2e_y,\sigma}c_{n,\sigma} + \text{h.c.} \right]. \tag{A.5}$$

To compute the transmissions in Fig. 6, we use a square lattice of size $L = 100$, with parameters $\mu = 0.5$, $\Delta = 0.1$, $t_x = 1.2$, $t_y = 0.8$, $t_{xy} = 0$, $t_{yy} = -0.5$, and phases $\phi_L = 2\pi/3$, $\phi_R = -2\pi/3$, and $\phi_T = 0$ for the left, right, and top regions respectively.

To compute the electrical conductance between a normal region and a Josephson junction in Fig. 7(a) we use a square lattice of size $L = 60$, and uniform parameters $\mu = 0.5$ and $t_x = t_y = 1$ for the whole system. In the superconducting regions we use $\Delta = 0.1$ and a phase difference $\phi_L - \phi_R = \pi/3$. To compute the electrical conductance between a normal region and a superconducting vortex in Fig. 7(b) we set up a three-dimensional system with additional nearest-neighbor $t_z$ hoppings in the z-direction and an onsite potential of $2t_z$. We use a lattice of size $L = 30$ and uniform parameters $\mu = 0.9$ and $t_x = t_y = t_z = 1$ for the whole system. In the superconducting region the superconducting phase forms a vortex with $\phi = \arctan(y/z)$, and the superconducting gap is position-dependent, $\Delta(y,z) = 0.25 \times \tanh(\sqrt{y^2 + z^2}/5)$.

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
