# Peer review of "Chiral adiabatic transmission protected by Fermi surface topology"

_SciPost Physics, doi:SciPost Phys. 18, 098 (2025)_

## Round 1 · Author Response

Response

We thank both referees for their valuable feedback and positive evaluation. We have taken their feedback into account and clarified multiple ambiguous or unclear details about the manuscript and its figures.

In the updated version we have also included a direct demonstration of the mechanism that protects the chiral adiabatic transmission. In the new Fig. 4, we confirm that the scattering wavefunctions do not overlap in phase space in the protected regime. When the separation breaks down, so does the quantized transmission. We also clarified the explanation of the mechanism along the lines of this demonstration.

We attach a redlined version of the manuscript with all the changes marked at the URL https://surfdrive.surf.nl/files/index.php/s/HWMAZOgyHajSHWw .

Referee 2

We thank the referee for their feedback and positive evaluation of the manuscript. Following their suggestions, we have improved the clarity of the manuscript by extending captions and adapting the figures. Below we list the detailed response to the referee suggestions.

  1. In Fig.2 (a), show the coordinate system x - y and a picture elucidating the definition of the angle \theta.

We have included the coordinate system and the definition of θ in Fig. 2(a) as suggested by the referee.

  1. In Fig. 3 in the inset, add the labels for the normal leads (1,2,3).

Taking into consideration both reports, we have changed Fig. 3 for a simpler illustration of the momentum expectation value of the scattering modes. In this figure, we have included the labels for the leads 1, 2, 3 which are consistent with the rest of the manuscript.

  1. In the insets of the left panel of Fig 4. the probability densities are not readable. I would suggest either to remove them or to show them in a separate figure together with an outline of the geometry.

Following the referee suggestion we have decided to remove the wavefunction densities. We have now also indicated that the geometries used for the calculations are those in a new Fig. 8. The Appendix includes an explanation on the parameters used as well.

  1. It took me some time to realise that in Figs 1-2 and 4-6, the results for states with energies outside the gap are depicted in a different colour (cyan). It would be really helpful to state this explicitly.

We thank the referee for pointing out this confusion. We have now changed the colors in all figures so that bulk bands are depicted in the same color as in-gap bands.

Referee 1

We thank the referee for their feedback on the manuscript, which we have incorporated to improve the clarity of the text and figures. A detailed response to the referee suggestions and questions are is listed below.

2- It is not clear to me whether a trijunction gives more information on the topology of the Fermi surface with respect to the single junction of Ref. [3].

We believe that the main focus of this research direction, both our work and Ref. 3, is not developing methods for probing the Fermi surface topology. This is better done using a tool like ARPES or quantum oscilations. Instead, we demonstrate how Fermi surface protects a range of phenomena. Our manuscript demonstrates one such phenomenon—chiral adiabatic transmission—which is distinct from the observations of Ref. 3. In the introduction we explain that it has no analogs.

1- The concept of energy barrier is unclear. It is related to Eq. (5), but more details are needed.

We thank the referee for bringing this up. The referee's request prompted us to carefully revisit and clarify our arguments. At its core, our argument relies on the separation of the different scattering wavefunctions in momentum and real space. We realized that this separation in the phase space is only guaranteed if all the trijunction angles are obtuse. As a consequence, our argument implies that in a trijunction with two acute angles, transmission should not be quantized. To verify this, we have computed the Wigner distribution (phase space probability distribution) of the scattering wave functions, and observed that:

  • In a trijunction with obtuse angles, all scattering wave functions are indeed separated in phase space.
  • In a trijunction with acute angles, some wave functions overlap in phase space.

In the updated manuscript, we present this result (new Fig. 4) as well as a reworded argument, which does not use a concept of an energy barrier, but rather describes which momentum orientations exist in each junction.

2- In Fig. 4: why the second panel is different from the right panel in Fig. 1? Why also T_{1,1} and T_{2,1} are different from 0? I was expecting only T_{3,1} to be quantized.

The system in Fig. 1 has equal phase differences of 2π/3 across each of the three arms. On the other hand, the system in Fig. 5 (previous Fig. 4) has unequal phase differences of 2π/3, 4π/3+0.5, 2π/3+0.5. This results in the Andreev states occupying different energy ranges in each arm (Fig. 5 left panel). At the energy where modes only exist in one arm, the only allowed process is a reflection of the Andreev modes. At the energy when a conduction channel opens in another arm, this changes to perfect transmission between two arms. Finally, only when the three arms have open channels, chiral transmission is possible and realized.

In the updated version of the manuscript we clarified in the caption that the phase differences are unequal, added a reference to the Appendix with parameter values from the main text, and explained the figure in the main text.

3- In Fig. 5, what is the meaning of unprotected transmissions (dashed lines)? What are the gray dashed lines?

In the previous version of the manuscript, the dashed lines meant to show the transmission eigenvalues that were not quantized. The gray color indicated the transmissions at the energies above the bulk gap.

To simplify the presentation, we made the line style and colors homogeneous.

4- Fig. 6 refers to a single Josephson junction? What are exactly T_{ea} and T_{ha}?

Figure 7(a) (previous 6(a)) shows the transmission from a single SNS junction into a wide normal region. Likewise, Fig. 7(b) (previous 6(b)) shows the transmission from a vortex core in a three-dimensional superconductor into a metallic lead. In the updated version of the manuscript we added the insets with the corresponding scattering geometries.

Tea is the transmission from Andreev modes into electrons, and Tha is the transmission of Andreev modes into holes. In the previous version of the manuscript this was defined in text, and we now included it in the caption. We thank the referee for point out this omission.

5- Many captions do not contain enough information to read the figures. For example: grey lines in Fig. 1 (right) and Fig. 5; values of k_x for the critical points in Fig. 2(b); ...

We thank the referee for bringing this omission up.

In the previous version of the manuscript, the gray lines indicated the transmissions above the bulk gap. In the updated version, we simplified all the figures and extended their captions to be more complete. We have also added the missing labels. Finally, we added kF to the x-axis in Fig. 2 as well.

---

## Round 1 · List of Changes

Figure changes:
- Homogenized linestyle and colors in Fig. 1
- Added theta to Fig. 2 and added kF to the x-axis
- Replaced Fig. 3
- Included a new Fig. 4 with Wigner distribution of scattering wavefunctions
- Removed wavefunction density insets from Fig. 5
- Added labels to Fig. 6 and homogenized linestyle and colors
- Included insets with the setups in Fig. 7 and labels to both panels
- Added new panels to Fig. 8 in the Appendix with the tight-binding models used in the main text

We attach a redlined version of the manuscript with all the text changes marked at the URL https://surfdrive.surf.nl/files/index.php/s/HWMAZOgyHajSHWw .

---

## Editorial Decision

published